# Oxidative Status of *Medicago truncatula* Seedlings after Inoculation with Rhizobacteria of the Genus *Pseudomonas*, *Paenibacillus* and *Sinorhizobium*

**DOI:** 10.3390/ijms24054781

**Published:** 2023-03-01

**Authors:** Anna Kisiel, Tymoteusz Miller

**Affiliations:** 1Institute of Marine and Environmental Sciences, University of Szczecin, Wąska 13, 71-415 Szczecin, Poland; 2Polish Society of Bioinformatics and Data Science BIODATA, Popiełuszki 4c, 71-214 Szczecin, Poland

**Keywords:** plant-growth-promoting rhizobacteria, reactive oxygen species, catalase, oxidative status in plant–microbe interaction

## Abstract

An increasing number of scientists working to raise agricultural productivity see the potential in the roots and the soil adjacent to them, together with a wealth of micro-organisms. The first mechanisms activated in the plant during any abiotic or biotic stress concern changes in the oxidative status of the plant. With this in mind, for the first time, an attempt was made to check whether the inoculation of seedlings of the model plant *Medicago truncatula* with rhizobacteria belonging to the genus *Pseudomonas* (*P. brassicacearum* KK5, *P. corrugata* KK7), *Paenibacillus borealis* KK4 and a symbiotic strain *Sinorhizobium meliloti* KK13 would change the oxidative status in the days following inoculation. Initially, an increase in H_2_O_2_ synthesis was observed, which led to an increase in the activity of antioxidant enzymes responsible for regulating hydrogen peroxide levels. The main enzyme involved in the reduction of H_2_O_2_ content in the roots was catalase. The observed changes indicate the possibility of using the applied rhizobacteria to induce processes related to plant resistance and thus to ensure protection against environmental stress factors. In the next stages, it seems reasonable to check whether the initial changes in the oxidative state affect the activation of other pathways related to plant immunity.

## 1. Introduction

Plant reactions leading to the activation of resistance mechanisms against pathogens are activated as a result of the action of signals called elicitors (biotic and abiotic) [1,2]. Biotic elicitors include fragments of cell walls and membranes and substances produced by both pathogenic and plant-growth-stimulating micro-organisms, including plant-growth-promoting rhizobacteria-PGPR [1,3,4,5]. These PGPRs promote plant growth, in particular by facilitating nutrient availability and modulating the host’s hormonal balance, but also show plant protective activity against pathogen entry [6,7,8,9,10,11]. Particularly interesting seems to be their importance for the activation of pathways leading to the activation of systemic immunity in plants [12,13,14,15].

Signaling pathways induced by microbial elicitor interactions generate various secondary messengers such as Ca^2+^, ROS, jasmonic acid, salicylic acid, and NO. The first mechanisms triggered in a plant during any abiotic or biotic stress relate to changes in the oxidative status of the plant. The reaction to contact with the elicitor is an oxidative burst, i.e., the release of reactive oxygen species (ROS) [16]. ROS include hydrogen peroxide (H_2_O_2_), superoxide anion radical (O^2−^), and hydroxyl radical (OH). They are produced in many basic biochemical processes in mitochondria, chloroplasts, peroxisomes and the apoplast [16,17,18]. The main source of ROS under conditions of abiotic and biotic stress is membrane NADPH oxidase (respiratory burst oxidase homolog) [17,19]. In Arabidopsis, 10 genes encoding this enzyme have been detected (AtRbohA-J) and in *Medicago truncatula* 7 (MtRbohA-G) [20].

Then, OH is converted by superoxide dismutase (SOD) into H_2_O_2_. The importance of H_2_O_2_ is enormous, with remarkable stability within cells (half-life of 10^−3^s). It can be transported by aquaporins, causing long-distance oxidative damage, and participate in cell signaling regulation, thus acting as elementary signaling molecules in the activation of plant immune responses [19,21,22]. The delicate balance between their production and scavenger activity enables the dual function of ROS in plants, maintained by a large network of enzymes and non-enzymatic antioxidant compounds [23]. The generated H_2_O_2_ is then converted to water and dioxygen by ascorbate peroxidase (APX) and glutathione peroxidase (GPX) and catalase (CAT) [22,23].

The plant response to abiotic or biotic stress usually turns out to be systemic and not limited to the place of ROS production [24]. In response to a pathogen attack, the release of ROS usually leads to the so-called hypersensitivity reaction (HR), which in turn leads to the death of tissues around the infected site and the activation of immune mechanisms in the plant and can lead to programmed cell death (PCD), which in turn is to prevent the spread of the pathogen [25,26]. ROS are not only considered as secondary messengers but activators of jasmonic acid (JA) that control gene expression including those responsible for the synthesis of secondary metabolites [27].

*Medicago truncatula* is considered an important model legume species, due to its small genome size, short life cycle, and autogenous reproduction [28,29,30,31] and it has been adopted as a model legume for genome sequencing and functional genomic research programs [32,33]. *M. truncatula* is the subject of many studies e.g., symbiosis with bacteria [34,35,36] and defense response to pathogens [37,38].

The aim of our study was to explore the impact of inoculation with free-living and symbiotic rhizobacteria on the oxidative status of *Medicago truncatula* plants, specifically focusing on the levels of hydrogen peroxide and the activity of hydrogen peroxide utilization enzymes in leaves and roots. It was hypothesized that inoculation with different strains of rhizobacteria will impact the oxidative status of the plants, as measured by changes in hydrogen peroxide levels and the activity of enzymes that utilize it. Therefore, *Medicago truncatula* plants were inoculated with free-living strains of bacteria belonging to the genus *Pseudomonas*, *Paenibacillus* and a symbiotic strain of the genus *Sinorhizobium*, and in the following days the effect of this inoculation on the level of hydrogen peroxide and the activity of antioxidant enzymes (CAT, APX) in leaves and roots was assessed. The test results may be useful for assessing the oxidative status during the interaction of plants with rhizobacteria. While there is some understanding of the potential benefits of using micro-organisms to enhance plant growth and productivity, there is still limited knowledge of the underlying mechanisms involved in the interaction between rhizobacteria and plants, particularly in relation to changes in oxidative status. This study aims to fill this knowledge gap and provide insights into the potential impact of rhizobacteria on plant oxidative status. 

## 2. Results

### 2.1. Effect of Rhizobacteria on Hydrogen Peroxide Production

Hydrogen peroxide (H_2_O_2_) was found before and after inoculation with rhizobacteria in both leaves and roots of *M. truncatula* seedlings.

In the five-week-old leaves and roots of the control seedlings, just before inoculation with the bacteria, an almost 30-fold higher content of hydrogen peroxide was found (72.2 µmol/g fresh weight) compared to the roots (2.5 µmol/g fresh weight). In non-inoculated seedlings, an increase in hydrogen peroxide content was observed over time; after 7 days, i.e., in 6-week-old seedlings compared to 5-week-old seedlings, its level more than doubled (162.7 µmol/g) in leaves and increased about 13-fold (31.9 µmol/g) in roots (Figure 1).

Inoculation with rhizobacteria affected the content of hydrogen peroxide in both leaves and roots of *M. truncatula* seedlings. Generally, the inoculated plants had lower levels of H_2_O_2_ (Figure 1). However, 24 h after inoculation, a decrease in the H_2_O_2_ content in the leaves of about 28% could be observed in the plants inoculated with *Paenibacillus borealis* KK4 and *Pseudomonas brassicacearum* KK5 and by 10% in plants inoculated with *Pseudomonas corrugata* KK7. Seedlings treated with *Sinorhizobium meliloti* KK13, after 24 h had higher levels of H_2_O_2_ (by 26%) than in the control leaves. After 3 and 7 days, the H_2_O_2_ level was lower in the leaves, similarly to the treatment with other bacteria, respectively by 9–29% on the third day and by 22–34% on the seventh day (Figure 1B).

In the roots of plants treated with rhizobacteria, the dynamics of hydrogen peroxide production had a different course (Figure 1A). After 24 h from inoculation, a significant increase in its content was found in the roots of seedlings inoculated with *P. brassicacearum* KK5, *P. corrugata* KK7 and *Sinorhizobium meliloti* KK13 by 78, 116 and 138%, respectively. The exception was the roots of seedlings treated with *Paenibacillus borealis* KK4, in which content decreased by 50%, and this downward trend was maintained at a similar level after 3 (57%) and 7 (66.7%) days. However, after 3 days in the roots of seedlings inoculated with *P. brassicacearum* KK5, *P. corrugata* KK7 and *Sinorhizobium meliloti* KK13, a sharp decrease in hydrogen peroxide content (from 37 to 57%) was observed, in contrast to 1 day. This decrease was maintained after 7 days, with the exception of the roots of seedlings inoculated with *Sinorhizobium meliloti* KK13, where its level was similar to that in the roots of control seedlings (Figure 1A).

Received results indicate that inoculation of 5-week-old *Medicago truncatula* seedlings with selected strains of bacteria modulates the level of H_2_O_2_. In the leaves of seedlings, a decrease in the content of hydrogen peroxide was observed from the first day (other than after treatment of the seedlings with a suspension of *Sinorhizobium meliloti* KK13—which saw an increase in the content of H_2_O_2_), which was maintained for 7 days. On the other hand, in the roots, which are the main contact point with the bacterial suspension, all the strains used for seedling inoculation caused an increase in the content of H_2_O_2_ after 24 h and a decrease in the following days.

The next step was to check whether the observed changes in hydrogen peroxide content in seedlings inoculated with bacteria could be related to the activity of enzymatic antioxidants, such as catalase and ascorbate peroxidase.

### 2.2. Effect of Rhizobacteria on the Activity of Oxidative Enzymes (CAT, APX)

Catalase activity in the leaves of 5-week-old *M. truncatula* seedlings was about 10 times higher than that in the roots of this plant (Figure 2). Changes in catalase activity in *M. truncatula* leaves and roots were observed over time. Within a week, a 1.5-fold decrease in the activity of this enzyme was observed in the leaves and 1.8-fold in the roots.

Inoculation of plants with rhizobacteria caused changes in catalase activity in both leaves and roots (Figure 2). On the first day after inoculation, the activity of this enzyme decreased in leaves of plants inoculated with *Pseudomonas brassicacearum* KK5, *Pseudomonas corrugata* KK7 and *Sinorhizobium meliloti* KK13 by 24, 17 and 28%, respectively. The exception was the leaves of seedlings treated with *Paenibacillus borealis* KK4, in which an increase in the activity of this enzyme by 11.8% was observed. Generally, 7 days after inoculation, catalase activity in the leaves of all inoculated seedlings was increased (by 50–65%) compared to the activity in control plants (Figure 2B).

On the first day, in the roots of plants inoculated with bacteria, the activity of this enzyme was reduced after inoculation of *Paenibacillus borealis* KK4, *Pseudomonas corrugata* KK7 and *Sinorhizobium meliloti* KK13, respectively by 42, 21 and 29% (Figure 2A). *P*. *brassicacearum* KK5 resulted in an 8% increase in root catalase activity. After 168 h from inoculation, all bacteria had a stimulating effect on catalase activity in the roots, with *Paenibacillus borealis* KK4 causing an increase of as much as 42%, both *Pseudomonas* strains by 23–26%, and *Sinorhizobium meliloti* KK13 by 10%.

In the five-week-old leaves and roots of the control seedlings, *M. truncatula* APX showed activity at a similar level, but in the leaves, it decreased twice over time (Figure 3B), while in the roots it remained at the level of approximately 176 µmol/min./mg protein (Figure 3A).

APX activity in leaves on the first day after treatment with *Paenibacillus borealis* KK4 and *Pseudomonas corrugata* KK7 decreases by 8.4% and 12%, while it increases under the influence of *Pseudomonas brassicacearum* KK5 and *Sinorhizobium meliloti* KK13 by 31% and 10%, respectively. On day 3, most of the strains caused a significant increase in leaf peroxidase activity in leaf (from 45% to 65%), with the exception of the *Paenibacillus borealis* KK4 strain, which still reduced the activity of this enzyme in leaves by 37%. On the seventh day, the activity of this enzyme increased after treatment with all bacterial strains from 62% to even 85% in the case of *Pseudomonas brassicacearum* KK5 (Figure 3B).

24 h after inoculation with bacteria in the roots of two strains, *Pseudomonas brassicacearum* KK5 and *Sinorhizobium meliloti* KK13 caused a slight (10% and 5%, respectively) increase in APX activity, and the next two strains, *Paenibacillus borealis* KK4 and *Pseudomonas corrugata* KK7, inhibited the activity of this enzyme by 20 and 50% (Figure 3A). On the third day after inoculation, strain KK4 still reduced peroxidase activity by 22%. The KK5 strain decreased APX activity by 13%, the other two strains caused an increase in the activity of this enzyme by 20% after treatment with KK7 and 11% after inoculation with KK13. On the 7th day after inoculation, strains of *Paenibacillus borealis* KK4 and *Sinorhizobium meliloti* KK13, induced it by 7% and 16%, respectively.

The analysis of variance showed that the factors defining the variability of enzyme concentrations in the tested parts of the plant could be the tested strains of rhizobacteria. The systems of highest importance were: strain, enzyme, plant part and their interaction. The strain-plant part system was characterized by the lowest significance index (Figure 4). 

The post-hoc test showed that significant relationships were noted between the leaf and the root—in particular, when it comes to the strains: control and strain KK13, strain KK4 and KK5, and strain KK4 and KK13 for all enzymes tested (Figure 4).

The analysis of clusters allowed for the identification of 3 significant groups—characterized by similar variations in the concentration of the studied enzymes. So, the first cluster shows aggregation mainly in the root between 0–24 h. The second cluster shows a similar variability between 24–72 h in both the leaf and the root of *M. truncatula*. The last cluster allowed for the identification of changes taking place in the leaf between 72–168 h (Figure 5).

The correlation analysis was visualized in Figure 6. In particular, a very strong negative correlation was observed in the leaf between H_2_O_2_ and APX, as well as between CAT and APX. A strong correlation was also noted between CAT and H_2_O_2_. In the case of the root, a moderate correlation was observed between the concentration of H_2_O_2_ and APX, and a strongly negative one between CAT and APX. Both organs showed a different level of interaction between the enzymes. The diverse interaction between APX and H_2_O_2_ is particularly noteworthy. Considering this, as well as the negative correlation between CAT and H_2_O_2_ in the root, it can be inferred that in the roots, CAT is mainly responsible for reducing the level of H_2_O_2_.

## 3. Discussion

The strains selected for the study were described as having the characteristics of PGPR (production of auxins, dissolution of phosphorus compounds, production of siderophores, the activity of ACC deaminase was checked) and as stimulators of *M. truncatula* growth (their effect on seedling weight was assessed) [39]. 

Inoculation of *M. truncatula* seedlings with free-living strains such as *Pseudomonas brassicacearum* KK5, *P. corrugata* KK7 and *Paenibacillus borealis* KK4 and the symbiotic strain *Sinorhizobium meliloti* KK13 caused changes in the oxidative status of the plant, measured by changes in the level of hydrogen peroxide and the activity of enzymes utilizing hydrogen peroxide, such as catalase and ascorbate peroxidase. Initially, the roots of *M. truncatula*, which is the contact point between the bacteria and the plant, showed an increase in the amount of H_2_O_2_. Taken together, these results indicate that the early seedling–rhizobacteria interaction is characterized by a strain-dependent H_2_O_2_ signature. The observed decrease in the content of H_2_O_2_ in leaves 7 days after the treatment of 5-week-old seedlings with suspensions of selected bacterial strains is associated with an increase in the activity of both catalase and ascorbate peroxidase. In the roots, however, catalase seems to be the main enzyme involved in the reduction of H_2_O_2_ content, because changes in the activity of ascorbate peroxidase were found only after inoculation of *M. truncatula* seedlings with *Paenibacillus borealis* KK4 and *Sinorhizobium meliloti* KK13 strains. Our study showed, for the first time, changes in the oxidative status of *M. truncatula* caused by inoculation with free-living PGPR.

Plants generally overcome threats from pathogenic micro-organisms through their innate ability to recognize signals from potential pathogens and then reprogram their defense systems to cope with those threats [40]. The rhizosphere microbiome plays a significant role in reprogramming plant defense responses [41,42,43,44]. The role of beneficial non-pathogenic microbes in stimulating the defense of host plants has been reported [45,46], suggesting that plant resistance is induced if successful interaction is achieved by non-pathogenic microbes [47,48,49,50].

Pieterse et al. firstly reported that systemic resistance induced by PGPR was independent of salicylic acid (SA) and pathogen-related (PR) proteins, but depended on jasmonic acid (JA) and ethylene (ET) signaling pathways. It was proposed to be the difference between induced systemic resistance (ISR) and systemic acquired resistance (SAR) [51]. However, many subsequent reports showed activation of both SA and JA/ET signaling pathways in rhizobacteria-induced ISR. This indicates the complexity and diversity of signaling pathways involved in ISR [52,53,54,55,56].

Each reaction of the plant organism begins with a local reaction at the point of exposure to elicitors. The first reaction upon contact with pathogenic and non-pathogenic micro-organisms is the disturbance of the oxidative status of the plant. Initially, plant–microbe interaction induces an oxidative burst locally with the release of reactive oxygen species, which also includes H_2_O_2_ [16,23,57,58]. This fact was also confirmed in our research, because 24 h after inoculation we observed an increase in the content of H_2_O_2_ (Figure 1). The induction of ROS is a significant signaling in control of various processes including immunity against pathogens, programmed cell death, and stomatal closure [59]. Owing to its relative stability compared to other ROS and its capacity for diffusing through aquaporins in the membranes and over more considerable distances within the cell, H_2_O_2_ acts as a stress signal transducer and contributes to numerous physiological functions in plants [60].

However, the accumulation of ROS also causes tissue cell damage [61]. Efficient capture of ROS by enzymatic and non-enzymatic reactions is required. Enzymatic mechanisms of ROS capture in plants are based, among others, on on peroxidase (POX), superoxide dismutase (SOD), ascorbate peroxidase (APX), glutathione peroxidase (GPX) and catalase (CAT) [22,23].

There are reports on the activity of antioxidant enzymes in *M. truncatula* during interaction with the symbiotic bacterium *S. meliloti* and their significant role during symbiosis. During nodulation, large amounts of reactive oxygen species are generated as a result of the continuous interaction between bacteria and plants [62]. Similarly, an increase in H_2_O_2_ content was observed in the roots of *M. truncatula* after inoculation with *S. meliloti* KK13 (Figure 1). Superoxide dismutases and catalases are considered essential for protecting the nitrogen fixation process in root nodules and are present in both symbiotic partners [62]. Mhadhbi et al. [63] after analyzing three lines of *M. truncatula* and three strains of *S. meliloti* showed a positive correlation between the number of nodules and the content of superoxide dismutase and the total protein content in nodules and the activity of catalase, while other antioxidant enzymes (glutathione and ascorbate peroxidase) did not show a clear correlation with the effectiveness of symbiosis. It is also known that during salt, osmotic or drought stress, reactive oxygen species are generated in huge amounts. Many researchers have proven that inoculation of *M. truncatula* seedlings with symbiotic bacteria protects plants against this type of stress by increasing the activity of antioxidant enzymes [64,65,66,67]. Kiirika et al. [68] showed, with the example of *M. truncatula* of the Jemalong line and plants with a transgene hindering the production of ROS, a high involvement of reactive oxygen species in interactions with both pathogenic (*Aphanomyces euteiches*) and symbiotic (*Glomus intraradices* and *Sinorhizobium meliloti*) micro-organisms. Despite the similarity between pathogenic and symbiotic interactions, the plant must retain the specificity of such responses that activate response signals and that are designed to promote or abolish the interaction. There are also reports on the beneficial effect of inoculation with free-living PGPR on the activity of antioxidant enzymes during salt or water stress in wheat, basil, rice, tomato and mint [69,70,71,72,73].

We also noticed that during the interaction of the plant with non-pathogenic micro-organisms the degradation of reactive oxygen species occurs (Figure 1). This is possible due to the increased activity of antioxidant enzymes, such as catalase (Figure 2), peroxidases (Figure 3). This can be an effective mechanism to attenuate the activation of plant defense responses and allow root colonization. It has previously been shown that ROS and antioxidant enzymes play a key role in the symbiosis between legumes and bacteria [74]. In tobacco colonized by the mycorrhizal fungus *Glomus mosseae*, a transient induction of catalase and ascorbate peroxidase activity was observed, possibly indicating the activation of defense responses and the release of free radicals in the early stages of symbiosis development [75]. ROS accumulation has been observed in *Medicago sativa* when forming a symbiosis with bacteria of the genus *Rhizobium* [76].

In this study, on the first day after inoculation with rhizobacteria, an increase in hydrogen peroxide content by up to 138% (except for *Paenibacillus borealis* KK4) was observed in the roots, which are the point of contact with rhizobacteria, and a decrease in this content in leaves by up to 28% (except for *Sinorhizobium meliloti* KK13). In the following days, the inhibitory effect of inoculation with all rhizobacteria on the production of hydrogen peroxide was clearly marked, which was especially visible on the 7th day in both leaves and roots of *M. truncatula* (Figure 1).

The level of hydrogen peroxide in the leaves and roots of *M. truncatula* seedlings inoculated with rhizobacteria was influenced by the activity of enzymatic antioxidants, such as catalase and peroxidase. The observed decrease in the production of H_2_O_2_ on the 7th day in the leaves was related to the increase in the activity of CAT (between 50 and 65%) and APX (between 62 and 85%) noted at that time, depending on the strain used. The main enzyme involved in lowering the H_2_O_2_ content in the roots is catalase; the highest increase in CAT activity was noted after treating the seedlings with a suspension of *Paenibacillus borealis* KK 4 and it was 42%, and APX only 7% (Figure 2 and Figure 3).

This study identified two major limitations that will need to be addressed in future studies. The main one is the lack of simultaneous co-inoculation with all the strains used in the study. This would allow us to observe the plant’s response and determine whether the strains act synergistically or antagonistically. However, this work was based on understanding the mechanism of the early interaction and determining the differential response depending on the strain. The co-inoculation effect will be important in constructing a consortium to prepare a vaccine for practical use. Another limitation may be the reaction of the strains themselves and the production of antioxidant enzymes by them, which can have a direct impact on the level of hydrogen peroxide in the plant.

Since the first report on PGPR-induced ISR in plants [77], much progress has been made in understanding the mechanisms for inducing and regulating plant defense responses. The plant defense system can be activated to resist various pathogen attacks and can also be suppressed to allow colonization of beneficial microbes. Both of these aspects of plant defense mediation work when the plant interacts with beneficial micro-organisms, and their interplay requires further study. The findings suggest that the tested strains can be used not only to stimulate plant growth, but also to stimulate their defenses. The study helped to elucidate the mechanisms underlying the early interactions between rhizobacteria and plants. This may be the basis for further research as it will allow for a better understanding of other processes activated in the plant as a result of contact with PGPR. The results also have practical implications as they will lead to more effective strategies for enhancing plant growth and controlling plant diseases by assessing early interactions. The big gap lies in how recognition of PGPR drives whole plants to improve growth and increase disease resistance. The first stage of this interaction is a change in the oxidative status of the plant, and this is when mechanisms are recognized and directed to defense or co-operation. Since there are already the first reports on the regulation of plant ROS by endophytes [78], this mechanism should be further explored and it should be checked whether the rhizobacteria used in our study directly contributed to the utilization of hydrogen peroxide in the plant. It also seems important to check whether the observed changes in oxidative status will translate into the activation of pathways related to plant immunity, such as the flavonoid biosynthesis pathway.

## 4. Materials and Methods

### 4.1. The Bacterial Strains

The studies used bacterial strains isolated from the *Medicago sativa* rhizosphere. They were free-living *Paenibacillus borealis* KK4, *Pseudomonas brassicacearum* KK5, *Pseudomonas corrugata* KK7 and the symbiotic *Sinorhizobium meliloti* KK13. The characteristics of the bacteria and their impact on the growth of *M. truncatula* were described in Kisiel and Kępczyńska [39].

### 4.2. Growth Room Pot Experiment

*Medicago truncatula* Gaertn ecotype Jemalong 5 (J5) was used in the research, whose seeds were obtained from the French National Institute for Agricultural Research (INRA) in France. The culture method was previously described in Kisiel and Kępczyńska [39]. The soil mixture containing sand and perlite (1:1, w/w) was sterilized. The seedlings were grown for 1 week under controlled growth room conditions with light–dark and temperature cycles (16 h light at 24 °C; 8 h dark at 20 °C). The light density was 150 lmol m^−2^s^−1^.

The one-week-old seedlings were inoculated with 10 mL bacteria inoculum (density of 10^8^ CFU ml^−1^) or 10 mM MgSO_4_. The inoculum of bacteria was prepared by growing bacterial cells in 20 mL of TSB and 2xYT medium, respectively, and incubated in a shaker incubator (200 rpm) at 28 °C. The cell suspension was diluted fivefold by the addition of sterile 10 mM MgSO_4_, and 10 mL portions of the dilution were used for seedling inoculation with a pipette, at a distance of 1 cm from the stem. The plants were irrigated with distilled water. The soil was fertilized once a week with N-low fertilizer (pH 5.8).

The oxidative status, i.e., the production of hydrogen peroxide and the activity of enzymes utilizing it (CAT, APX), were analyzed in the roots and leaves of *M. truncatula* J5. The material for the analysis consisted of roots and leaves of 5-week-old seedlings treated with rhizobacteria, collected at T_0_ (at the time of treatment with bacteria) and after 24, 72 and 168 h after treatment. Roots and leaves of seedlings treated with 10 mM MgSO_4_ served as control.

### 4.3. Determination of H_2_O_2_ Content

Analysis of the content of H_2_O_2_ in plant tissues was carried out using the spectrophotometric method according to Velikova et al. [79]. Plant tissue (ca. 0.1 g) was pounded in a mortar and mortar placed on ice, 1 mL of 0.1% TCA was added and pounded for ca. 3 min. The extract was poured into an Eppendorf tube and centrifuged (4 °C/15 min./12,000 xg). To the 750 μL supernatant was added 750 μL of 0.1 M phosphate buffer pH 7.0 and 1.5 mL of 1 M KI. The reaction mixture was incubated in a heating block at 30 °C for 5 min. and poured into a 3 mL quartz cuvette. Absorbance was measured on a spectrophotometer at 390 nm. The results were recalculated according to the standard curve prepared for known concentrations of H_2_O_2_ and expressed in µmol·g^−1^ fresh weight.

### 4.4. Determination of Catalase Activity

In order to analyze the activity of enzymatic antioxidants (CAT and APX), it was necessary to obtain a plant extract. So, 0.1 g of tissue (leaves/roots) was placed in a chilled mortar and homogenized in liquid nitrogen. Then, 1.5 mL of extraction buffer composed of 0.1 M phosphate buffer pH 7.0, 1% PVP and 1 mM EDTA was added. The finished extract was poured into a centrifuge tube and centrifuged (4 °C/20 min/15,000× *g*). The extract was then transferred to new tubes and used for CAT and APX assays.

Catalase activity was determined by the method described by Mhadhbi et al. [63] with modifications, measuring the decrease in absorbance at λ = 240 nm. A reaction mixture containing 2.7 mL of buffered hydrogen peroxide (0.036% H_2_O_2_ in 10 mM potassium phosphate buffer, pH 7.0) and 300 μL of supernatant was prepared in a quartz cuvette. The contents of the cuvette were stirred rapidly and the reaction kinetics were measured by observing the decrease in absorbance over 1 min using a spectrophotometer. Catalase activity was determined on the basis of a standard curve for known concentrations of H_2_O_2_ and expressed in µmol H_2_O_2_ min^−1^ mg^−1^ protein.

### 4.5. Determination of Ascorbate Peroxidase Activity

Simultaneously, the leaf and root extract supernatant (as described above) were used to determine APX activity by the method described by Mhadhbi et al. [63] using a spectrophotometer, measuring the decrease in absorbance at λ = 290 nm, which occurs as a result of the oxidation of sodium ascorbate to dehydroascorbate.

The reaction mixture was prepared in 5 mL glass tubes. To 100 μL of the supernatant was added 2.5 mL of 0.1 M potassium phosphate buffer pH 7.0 and 200 μL of 5 mM sodium ascorbate. The reaction mixture was incubated in a water bath at 35 °C for 5 min. Measurements were made in 3 mL quartz cuvettes into which the reaction mixture was poured. The reaction was initiated by adding 200 μL of 1 mM H_2_O_2_ to the cuvette. Enzyme activity was determined based on the decrease in absorbance in 1 min. after the start of the reaction. The results were converted according to the calibration curve for sodium ascorbate. Enzyme activity was expressed in µmol of sodium ascorbate·min^−1^·mg^−1^ of protein, which means the ability of the enzyme to convert 1µmol of ascorbate to dehydroascorbate in 1 min.

### 4.6. Protein Content Determination

The protein content of the enzyme extract was determined by the dye-binding method using the Bradford reagent (Sigma-Aldrich) and bovine serum albumin (Sigma-Aldrich) as standard [80].

### 4.7. Statistical Analysis

To elucidate the impact of selected rhizobacterial strains on the activity of H_2_O_2_, CAT, and AXP in the leaves and roots of *Medicago truncatula*, we conducted a comprehensive set of analyses, including: (1) Pearson correlation analysis, represented using heatmap visualization; (2) Analysis of variance (ANOVA) with post-hoc Tukey’s honestly significant difference test (HSD); and (3) Agglomerative hierarchical cluster analysis with Ward’s linkage and square Euclidean distance, with Sneath’s criterion employed to assess cluster significance. To aid in result interpretation, we performed a pair grid for plotting pairwise relationships within the dataset. 

All statistical analyses were executed using Python version 3.9.13 in PyCharm Professional IDE (jetbrains.com). The publicly accessible repository on GitHub (https://github.com/PTBDBIODATA accessed on 1 January 2023) contains the database, supplementary materials, requirements, and source code.

## 5. Conclusions

This study discusses the early plant response to interaction with rhizobacteria. It was noted that the first reaction is an increase in the content of hydrogen peroxide, especially visible in the roots of *M. truncatula*, 24 h after inoculation with rhizobacteria. This is particularly important due to its signal nature in many processes. In the following days, the activity of enzymes decomposing hydrogen peroxide, i.e., catalase and ascorbate peroxidase, increased. This confirms the possibility of stimulating the production of hydrogen peroxide, a known inducer of immune processes, and controlling its level by enzymatic antioxidants (catalase and peroxidase) using PGPR. Changes in oxidative status are one of the first steps that can lead to plant defense. The use of beneficial micro-organisms to induce plant defense responses against pathogenic micro-organisms will be an environmentally friendly alternative to dangerous chemical pesticides in disease control. Only a thorough understanding of the mechanisms triggered by individual strains of rhizobacteria on a plant model, such as *Medicago truncatula*, will allow the creation of comprehensively acting biopreparations.

## Figures and Tables

**Figure 1 ijms-24-04781-f001:**
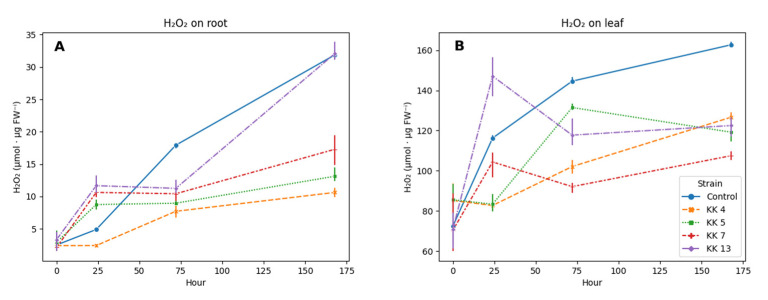
Influence of selected rhizobacteria on hydrogen peroxide content in root (**A**) and leaves (**B**) of 5-week-old seedlings of *M. truncatula* J5 and after 24, 72 and 168 h after inoculation.

**Figure 2 ijms-24-04781-f002:**
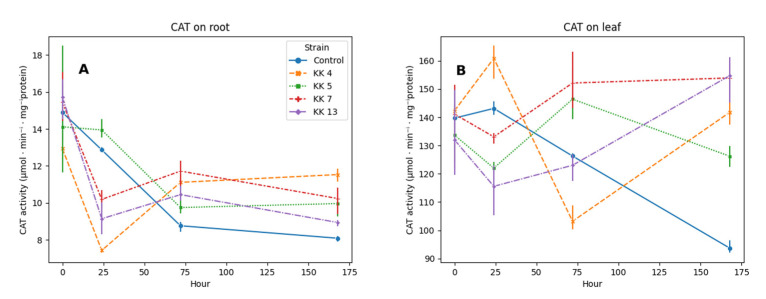
Effect of selected rhizobacteria on catalase activity (CAT, EC 1.11.1.6) in root (**A**) and leaves (**B**) of 5-week-old *M. truncatula* J5 seedlings at 24, 72 and 168 h after inoculation.

**Figure 3 ijms-24-04781-f003:**
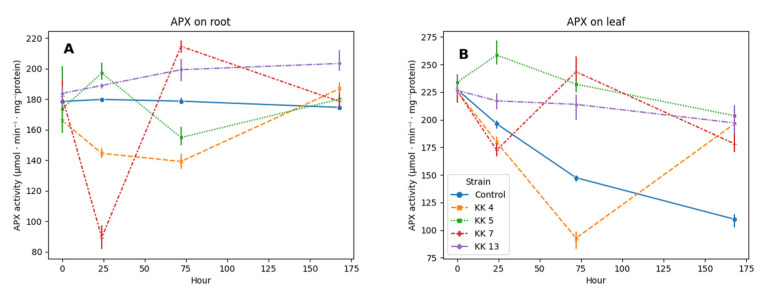
Effect of selected rhizobacteria on catalase activity (APX, EC 1.11.1.11) in root (**A**) and leaves (**B**) of 5-week-old seedlings of *M. truncatula* J5 at 24, 72 and 168 h after inoculation.

**Figure 4 ijms-24-04781-f004:**
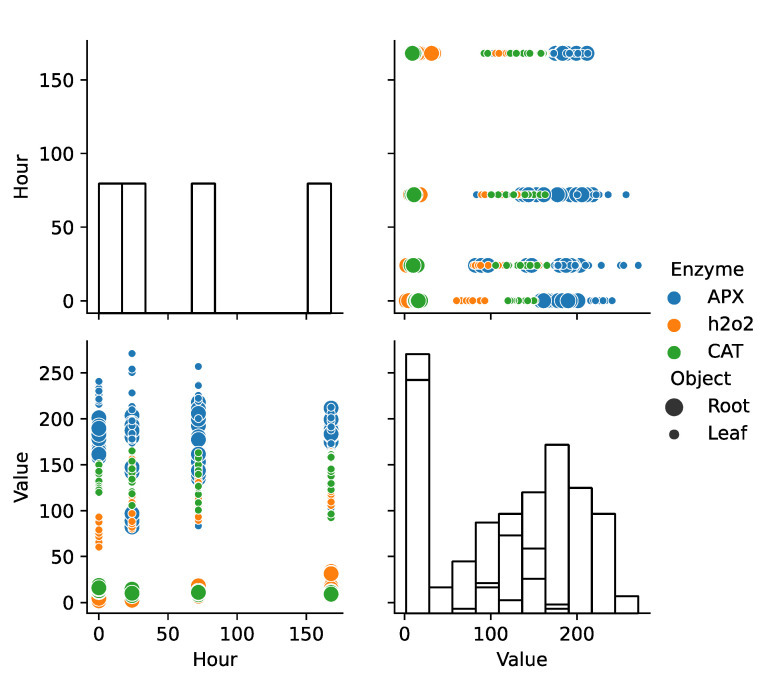
Quantitative analysis of obtained results—pairwise relationships.

**Figure 5 ijms-24-04781-f005:**
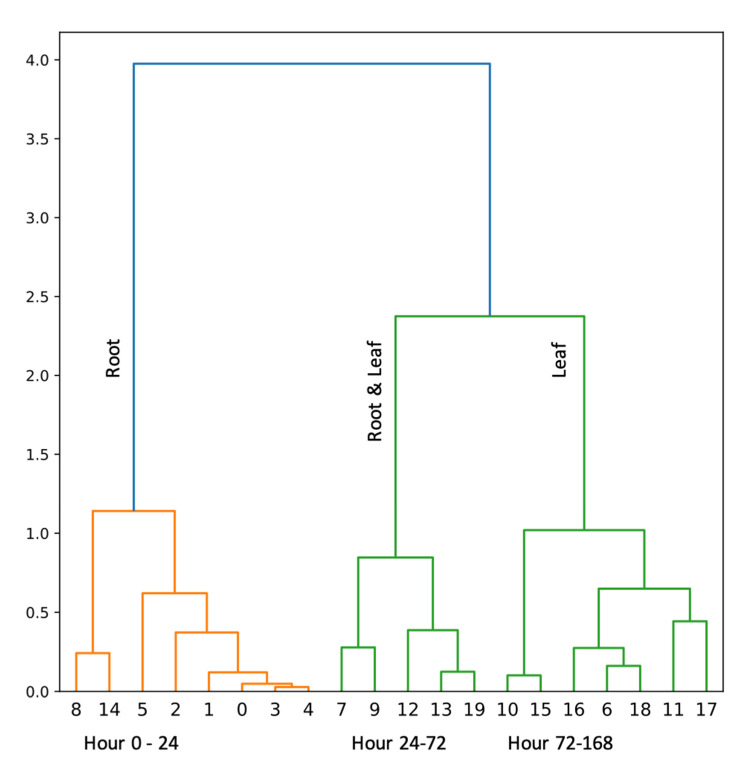
Cluster analysis results—similarities between changes in the concentration of tested enzymes in the leaf and root of *Medicago truncatula* at different hours of the experiment.

**Figure 6 ijms-24-04781-f006:**
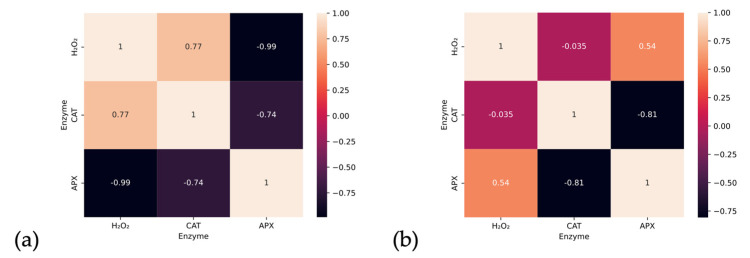
Correlation between studied enzymes in leaf (**a**) and root (**b**).

## Data Availability

The database, supplementary materials, requirements and source code are stored in public repository on the GitHub (https://github.com/PTBDBIODATA accessed on 1 January 2023).

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
