# Peer review of "Oxidative Status of *Medicago truncatula* Seedlings after Inoculation with Rhizobacteria of the Genus *Pseudomonas*, *Paenibacillus* and *Sinorhizobium"

_ijms, 2023, doi:10.3390/ijms24054781_

Round 1
Reviewer 1 Report
Medicago truncatula is an important model plant species, and it has been used as a model legume for the sequencing of genomes as well as functional genomic research programs. Many studies have been done on M. truncatula, including ones on symbiosis with bacteria and the plant's defense response to pathogens. The author attempts to explain the mechanism by which free-living and symbiotic rhizobacteria interact with plants. The topic is interesting, and the results are good. Some comments or suggestions for consideration to improve the quality of this manuscript.
Abstract
The abstract looks like too long. I am not sure whether it meets the requirements of the journal. My major concern is that many basic background information, and some colloquial sentences/phrase were shown in this part. For example, lines 10-11, the information of green revolution. Line 25, in the following days…In the abstract section, the authors should focus on your own research and show the highlights of this study. and based on this, give us the conclusions or implications.
Introduction
Line 63,64: “Generated H2O2 is then converted to water and dioxygen by ascorbate peroxidase (APX) and glutathione peroxidase (GPX), and catalase (CAT)” should be replace like this “The generated H2O2 is then converted to water and dioxygen by ascorbate peroxidase (APX), glutathione peroxidase (GPX), and catalase (CAT).”.
7) Line 79–85: Should be more clear, and the purpose of the research should be made more clear, as well.
Results
Generally, this sections is good.
Discussion
Line 202: “pathogenic microorganisms by their innate ability” It would be more appropriate to use the word "through" in place of "by."The sentence should be like this “pathogenic microorganisms through their innate ability”.
Lines 269–272: Please modify this sentence, and make it more concise and clear.
Conclusion
In line 358, this review?
In this section, the authors should pay more attention on your own study, summarize your results and give us some home message. Please add some information in this section.
References
13) Some references have small errors. For example, commas, dots, and other punctuation marks should be corrected and checked all the citations and references according to journal instructions and guidelines.
Figures
For figures 1-3, please re-draw the lines in these figures, make them more clear and easy to separate each other.
Author Response
Thank You for all the tips and comments. We appreciate the amount of work you put into the review. I hope that we have managed to apply all the comments and thanks to this our work has become more precise and transparent.
Below we present the most important changes.
Abstract has been shortened and changed. General information has been omitted and the focus has been on own research and conclusions.
Introduction was modified according to comments:
Line 63,64- changed in lines 78-80,
Line 79-85 - changed in lines 94-110.
Discussion was supplemented and modified in order to give the results a broader context and take into account the possibility of their further application. The biggest changes can be observed in the lines 242-261, lines 276-272, lines 285-303, lines 367- 396.
Line 202- changed in line 269.
Lines 269–272 modified in lines 232-240.
Conclusion: This section has been significantly modified -lines 495-508.
References and Figures have been corrected.
Thank You for all your Time. We hope that our addons to manuscript fulfill Yours expections.
Reviewer 2 Report
Title
The title suggests that the paper investigates the oxidative status of Medicago truncatula after inoculation with two types of Plant Growth-Promoting Rhizobacteria (PGPR). While investigating the effect of PGPR on the oxidative status of plants is a relevant research topic, the title does not clearly indicate what is new or unique about this particular study. To improve the novelty of the title, the authors could consider adding more specific information about the type of PGPR used, the method of inoculation, or the specific aspects of oxidative stress that are being investigated.
Abstract
In terms of the problem statement, while the abstract does touch on the importance of understanding the mechanisms of plant-microbe interactions for developing environmentally friendly solutions for agriculture, it does not provide a clear and specific research question or hypothesis that the study aims to address. Additionally, the language used in the abstract is somewhat vague and imprecise, which can make it difficult for readers to fully understand the goals and significance of the research. In terms of the conclusion, the abstract provides a very brief summary of the findings but does not provide any interpretation or discussion of their significance. Similarly, there are no future prospects or suggestions for further research provided in the abstract. Overall, while the abstract does provide some information about the study, it could be improved by providing more specific details about the research question, methodology, and findings, as well as more thorough interpretation and discussion of the results and their significance.
Introduction
Please incorporate that at the end of introduction. Authors can modified my suggestions as per their understanding.
Novelty statement: The study aims to explore the impact of inoculation with free-living and symbiotic rhizobacteria on the oxidative status of Medicago truncatula plants, specifically focusing on the levels of hydrogen peroxide and the activity of hydrogen peroxide utilization enzymes in leaves and roots.
Hypothesis: The researchers hypothesize that inoculation with different strains of rhizobacteria will impact the oxidative status of the plants, as measured by changes in hydrogen peroxide levels and the activity of hydrogen peroxide utilization enzymes.
Knowledge gap: While there is some understanding of the potential benefits of using microorganisms to enhance plant growth and productivity, there is still limited knowledge of the underlying mechanisms involved in the interaction between rhizobacteria and plants, particularly in relation to changes in oxidative status. This study aims to fill this knowledge gap and provide insights into the potential impact of rhizobacteria on plant oxidative status.
Material and Methods
Statistical analysis: Please modified it as “To elucidate the impact of selected rhizobacterial strains on the activity of H2O2, CAT, and AXP in the leaves and roots of Medicago truncatula, we conducted a comprehensive set of analyses, including: 1) Pearson correlation analysis, represented using heatmap visualization; 2) analysis of variance (ANOVA) with post-hoc Tukey Honestly Significant Difference test (HSD); and 3) Agglomerative Hierarchical Cluster Analysis with Ward's linkage and Square Euclidean distance, with Sneath's Criterion employed to assess cluster significance. To aid in result interpretation, we performed a Pair Grid for plotting pairwise relationships within the dataset. All statistical analyses were executed using Python version 3.9.13 in PyCharm Professional IDE (jetbrains.com). The publicly accessible repository on GitHub (https://github.com/PTBDBIODATA) contains the database, supplementary materials, requirements, and source code”
Results
Please follow suggestion provided below while writing the results
1. Use more concise and precise language: Instead of saying "In general, there was always less of it," you could say "Generally, the inoculated plants had lower levels of H2O2."
2. Use active voice: Instead of saying "The exception was the leaves of seedlings treated with Sinorhizobium meliloti KK13," you could say "Seedlings treated with Sinorhizobium meliloti KK13 had higher levels of H2O2."
3. Use consistent verb tense: In the paragraph, both past and present tense are used. Try to use past tense consistently for describing the experiment and its results.
4. Use proper formatting: Use proper punctuation, capitalization, and italics for scientific names and symbols, e.g. "M. truncatula," "Pseudomonas borealis KK4," and "H2O2."
5. Provide more context: It may be helpful to provide more context on why the changes in H2O2 levels are significant and what they indicate about the interaction between the plants and rhizobacteria.
Discussion
6. Start by summarizing the main findings of the study and how they relate to the research question and hypothesis.
7. Provide a context for the results by discussing previous research on the topic and how the current study builds on or challenges existing knowledge.
8. Explore the potential mechanisms underlying the observed effects of the rhizobacteria on hydrogen peroxide levels and enzyme activity. This could include a discussion of how the bacteria might be interacting with plant defense pathways, such as through the production of phytohormones or by inducing systemic acquired resistance.
9. Address any limitations of the study and potential sources of error or bias, and discuss how these limitations might affect the interpretation of the results.
10. Consider the broader implications of the findings and how they might be applied in future research or practical applications, such as in the development of more effective strategies for enhancing plant growth or controlling plant diseases.
11. Finally, identify any remaining knowledge gaps or areas for future research, and suggest potential directions for further study.
12. By incorporating these elements into the discussion, you can help provide a clear and comprehensive understanding of the study's findings, their implications, and their potential for advancing our knowledge of plant-microbe interactions.
Author Response
Thank You for all the tips and comments. We appreciate the amount of work you put into the review. I hope that we have managed to apply all the comments and thanks to this our work has become more precise and transparent.
Below we present the most important changes.
Title has been changed to more accurately reflect the research described in this manuscript
Abstract has been shortened and changed. General information has been omitted and the focus has been on own research and conclusions.
Introduction changed and reviewer's suggestions included (lines 94-110).
Material and Methods: Statistical analysis was changed (lines 470-481).
Results:
An attempt was made to improve the language.
Provide more context: It may be helpful to provide more context on why the changes in H2O2 levels are significant and what they indicate about the interaction between the plants and rhizobacteria- added lines 246-261 and lines 285-294).
Discussion:
The discussion was modified and supplemented based on the reviewer's comments to give the results a broader context and take into account the possibility of their further application. The biggest changes can be observed in the lines 242-261, lines 276-272, lines 285-303, lines 367- 396.
Conclusion
This section has been significantly modified -lines 495-508.
Thank You for all your Time. We hope that our addons to manuscript fulfill Yours expections.
Author Response
Thank You for all the tips and comments. We appreciate the amount of work you put into the review. I hope that we have managed to apply all the comments and thanks to this our work has become more precise and transparent.
Below we present the most important changes.
Abstract has been shortened and changed. General information has been omitted and the focus has been on own research and conclusions.
Introduction
Line 80 to 85 - changed in lines 94-110.
Results
Line 150 and Line 156 lines have been corrected and applied for figures 2 and 3.
The authors did not present any comments on the results in relation to figure 6 (Figure 6. Correlation between studied enzymes in leaf (a) and root (b) - added on lines 232-240.
Discussion was supplemented and modified in order to give the results a broader context and take into account the possibility of their further application. The biggest changes can be observed in the lines 242-261, lines 276-272, lines 285-303, lines 367- 396.
Conclusions This section has been significantly modified -lines 495-508.
Thank You for all your Time. We hope that our addons to manuscript fulfill Yours expections.
Reviewer 4 Report
In the work by Kiesel and Miller, they analyze the oxidative response of M. truncatula plants to the inoculation of some bacterial strains. However, it seems to me that the work is based on a single experiment that does not show anything new or any contribution to the field. Some graphs are based on the analysis of the same data, being redundant.
First, the bacteria, although they provide a reference to their plant growth-promoting capacities, the authors do not show graphs of this effect in plants, nor of single or consortium inoculation.
It seems to me that this experiment is essential to correlate the promoter effect with the oxidative response of the plant to the inoculation with each strain.
Another suggestion is to trim the justification of the Abstract, it is too long.
They should also test the co-inoculation of the mixture of all the strains and see a response in the plant to the interaction of the consortium. So one wonders if the strains are antagonistic or synergistic with each other. and what is the response of the plant?
In my opinion, I have seen more complete papers in the IJMS journal and this manuscript does not show innovative data that justifies its publication.
Author Response
Thank You for all the tips and comments. We appreciate the amount of work you put into the review. I hope that we have managed to apply all the comments and thanks to this our work has become more precise and transparent.
The results presented in the manuscript are the fruit of many experiments - and are a novelty that has not been published so far, which equally proves their scientific value. The graphs are not redundant - and they are not based on a single measurement series.
The aim of the work was not to illustrate the known promotional effect of growth on plants. However, such work has already been published and we have added this information to the manuscript (lines 242-245).
The abstract has been shortened and changes. General information has been omitted and the focus has been on own research and conclusions.
Studying consortia and bacterial interdependencies are part of ongoing research. At the moment - we do not want to publish these studies. Information about this has been added to lines 367-375.
Discussion was supplemented and modified in order to give the results a broader context and take into account the possibility of their further application. The biggest changes can be observed in the lines 242-261, lines 276-272, lines 285-303, lines 367- 396.
Thank You for all your Time. We hope that our addons to manuscript fulfill Yours expections.
Round 2
Reviewer 2 Report
Dear Authors,
I appreciate the effort you have put into this manuscript and believe it will make a valuable addition to our publication. We look forward to sharing your work with our readers and colleagues in the field.
Thank you for choosing our journal as a platform to share your research findings. We hope to hear more from you in the future.
Best regards,
Reviewer 4 Report
Authors addressed all suggestions